# A Simulation Study of a Gate-All-Around Nanowire Transistor with a Core–Insulator

**DOI:** 10.3390/mi11020223

**Published:** 2020-02-21

**Authors:** Yannan Zhang, Ke Han, Jiawei Li

**Affiliations:** School of Electronic Engineering, Beijing University of Posts and Telecommunications, Haidian district, Beijing 100876, Chinahanke@bupt.edu.cn (K.H.)

**Keywords:** CMOS, core-insulator, gate-all-around, field effect transistor, GAA, nanowire

## Abstract

Ultra-low power and high-performance logical devices have been the driving force for the continued scaling of complementary metal oxide semiconductor field effect transistors which greatly enable electronic devices such as smart phones to be energy-efficient and portable. In the pursuit of smaller and faster devices, researchers and scientists have worked out a number of ways to further lower the leaking current of MOSFETs (Metal oxide semiconductor field effect transistor). Nanowire structure is now regarded as a promising candidate of future generation of logical devices due to its ultra-low off-state leaking current compares to FinFET. However, the potential of nanowire in terms of off-state current has not been fully discovered. In this article, a novel Core–Insulator Gate-All-Around (CIGAA) nanowire has been proposed, investigated, and simulated comprehensively and systematically based on 3D numerical simulation. Comparisons are carried out between GAA and CIGAA. The new CIGAA structure exhibits low off-state current compares to that of GAA, making it a suitable candidate of future low-power and energy-efficient devices.

## 1. Introduction

Ultra-low power and high-performance logical devices have been the driving force for the continued scaling of complementary metal oxide semiconductor field effect transistors which greatly enable electronic devices such as smart phones to be energy-efficient and portable, while system scaling enabled by the Moore’s law is facing challenges due to the scarcity of resources such as power and interconnect bandwidth. Constrained by the limited capacity of battery, portable electronic devices are hard to have an “always-on” feature and have to be recharged frequently, causing great inconvenience to users. To reduce the overall power consumption, researchers have worked out a number of ways to reduce the off-state current of CMOS devices [1]. By fabricating the whole system on anSOI (Silicon on insulator) wafer, the stand-by current of the system can greatly reduce due to the low off-state current of SOI MOSFET [2]. However, the SOI MOSFET has self-heating effect due to poor heat conductivity of buried silicon dioxide layer (BOX) which increases device operating temperature, reduces carrier mobility as well as causes performance degradation [3,4]. By introducing new physical mechanics into CMOS devices, researchers are able to lower the subthreshold slope of transistors and hence reduce the leaking current of whole system. These types of devices include Impact Ionization MOS (IMOS) [5] and Tunnel Field Effect Transistors (TFET) [6]. Technically, IMOS is a reverse biased p-i-n diode with a control gate. The control gate is used to control impact ionization phenomenon between two junctions. The avalanche breakdown is a very fast and gated diode pulsed into breakdown can show subthreshold slopes much lower than 60 mV/dec [7], and thus exhibits lower off-state current compared with a conventional MOSFET [8]. However, due to the need of drastic doping profile, the fabrication of IMOS requires costly millisecond annealing techniques which greatly limits its application [9]. A tunnel field effect transistor is designed using the band-to-band tunneling effect. The carriers are injected by a band-to-band tunneling effect from a valence band of source for a N-type TFET, which is totally different from conventional CMOS devices that use thermionic emission [10]. The physical mechanics of TFET allow them not to be constrained by the Boltzmann limit (about 60 mV/dec at room temperature). Thus, TFET has the potential to be used as low-power devices for its extremely low off-state current. However, the TFETs fabricated are not competitive with conventional MOSFETs which are based on thermionic emission. Low on-state current and high average subthreshold slope (V${}_{g}$-dependent subthreshold slope) are main limitations of TFET devices [11,12]. Gate-All-Around (GAA) CMOS FET is based on conventional CMOS FET; it features a circular gate around the channel. GAA MOSFET is compatible with an existing CMOS fabrication process; it has the superior electrostatic control compared with FinFET and planar CMOS FET. The ITRS predicted that, beyond 2020 [1], a transition to Gate-All-Around and vertical nanowires devices will be needed when there will be no room left for the scaling because GAA devices are the ultimate structure in terms of electrostatic control to scale to the shortest possible effective channel length. While we found the potential of GAA devices has not been fully discovered, by introducing a core-Insulator into conventional GAA devices (we called it a Core–Insulator Gate-All-Around nanowire), the off-state current is expected to be further lowered, which makes it more suitable for fabricating low-power devices. The introduction of a Core–Insulator does not have any exotic materials, so it is highly compatible with a current fabrication process. Our experiments show that, because of the presence of Core–Insulator, the off-state current is lowered by more than two times, and it shows better subthreshold characteristics. We believe that this newly proposed structure can be a promising candidate of future low-power and energy-efficient CMOS devices.

## 2. Device Structure and Experiment Methodology

### 2.1. Descriptions of CIGAA Structure

The difference between conventional GAA and CIGAA (Core-Insulator Gate-All-Around) is that a CIGAA structure has a Core–Insulator between the channel, as shown in Figure 1. The material of Core–Insulator can be SiO_2_, Si_3_N_4_ and so on, and the impact of different material of Core–Insulator will be addressed in the following paragraph. We use HfO_2_ as a gate dielectric because it has a low leaking current and high dielectric constant, which can greatly improve the performance of the device without increasing the gate leaking current. The channel of CIGAA structure is not a solid cylinder as that of conventional GAA structure; it is a tubular channel. The gate metal should be carefully selected to tune the work function for a particular threshold voltage requirement.

### 2.2. Simulation Physical Models

Our simulation platform is Sentaurus TCAD 2017 Version N-2017.09 [13]. To describe the current densities of electrons and holes, we introduced a Drift–Diffusion [14] model that takes into account the contribution of electron affinity, the band gap as well as the spatial variations [15,16] of the electrostatic potential. Because the oxide thickness and channel width have reached quantum-mechanical length scales, the wave nature of electrons and holes can no longer be neglected, thus Density-Gradient [17] is used to simulate quantization effects. In order to describe the effects of electron–hole scattering, the screening of ionized impurities by charge carriers, and the clustering of impurities, Philips Unified Mobility [18] is used. Since HfO_2_/Silicon interface can lead to a mobility degradation [19,20], we also must take this into consideration by including a Lombardi Mobility Degradation model [21]. Hurkx Trap Assisted Tunneling models [22,23,24] are incorporated to simulate the tunneling effects at such small dimensions. In addition, a quantum potential model [25] was also taken into consideration. Because the source and drain are highly doped, we use a band gap narrowing model [26] to simulate this effect.

### 2.3. Structure Parameters Used for Simulation

All the parameters of our experiment are shown in Table 1 and Figure 2. Both structures have the same diameter as well as doping profiles. The source/drain doping concentration is $1\times 10^{20}$ atoms/cm^3^. Channel is lightly doped, which is $1\times 10^{15}$ atoms/cm^3^. For this article, channel length is fixed to 15 nm and the length of drain and source are both fixed to 10 nm. The diameter of Core–Insulator is set from 2.0 nm to 14 nm. The gate dielectric is HfO_2_, and the thickness is shown in a table. For comparison, we have also simulated a conventional GAA nanowire of the same overall dimensions.

### 2.4. Considerations of Workfunction

Ideally, conventional GAA and CIGAA will have different threshold voltage although they have same geometric parameters, since the presence of Core–Insulator will affect the device threshold voltage. In order to better illustrate and compare the performance of two structures, in other words, to have a fair comparison, we must tune their gate workfunction to ensure they have same threshold voltage, so that we can compare their performance by the same benchmark. It is noteworthy that it is hard to tune the workfunction at any desired value in the real fabrication process, although it can be easily achieved in TCAD simulation; all we want is to compare the performance difference between CIGAA and GAA under TCAD simulation.

### 2.5. Suggested Fabrication Process Flow for CIGAA

Based on the previous works of nanotube MOSFETs [27,28,29], the suggested fabrication process flow for CIGAA is shown in Figure 3. The CIGAA can be realized using the process flow suggested in [28] with some major changes. The first steps of the fabrication process is to form a cylindrical-shaped outer silicon layer with a sidewall using electron beam lithography (EBL) and sidewall deposition (Figure 3a–e), as suggested in [28]. Then, the source side spacer is formed using low-pressure chemical vapor deposition (LPCVD), and the following step (Figure 3f) is to remove unnecessary spacer material that is covered on the sidewall and cylindrical-shaped outer silicon layer using lithography and etching. Subsequently, gate oxide should be formed; the first is to deposit a thin layer HfO_2_ using low-pressure chemical vapor deposition (LPCVD). To remove redundant HfO_2_, lithography is used to protect gate oxide and anisotropic etching to remove unnecessary HfO_2_, as shown in Figure 3g–i. The next step is to form and partial removal of gate metal, as shown in Figure 3j,k. Subsequently, a sacrificial layer surrounding the top spacer is deposited using low-pressure chemical vapor deposition (LPCVD) and chemical mechanical polishing (CMP), which is shown in Figure 3l,m. Using selective etching followed by deposition of nitride and removal of the sacrificial layer, as shown in Figure 3n–p, the structure is prepared to form Core–Insulator. By anisotropic etching of silicon, the channel is formed, as shown in Figure 3q. The following step is to deposit Core–Insulator; this is illustrated in Figure 3r. Finally, the drain side is deposited with silicon and spacer, and the contacts are formed to finalized the device, as shown in Figure 3s–u.

## 3. Results and Discussions

### 3.1. Basic Characteristics of CIGAA and GAA

Figure 4a shows the result of on-state current (I${}_{on}$). The on-state current of both structures increase linearly when channel thickness (T${}_{ch}$) increases, since the increment of channel thickness means that the effective width of channel will also increase. When T${}_{ox}$ is same, CIGAA exhibits a slightly lowered on-state current compared with that of GAA. The on-state current degradation of CIGAA is due to the reduction of the total volume of channel because of the presence of Core–Insulator, which results in smaller effective channel width. However, the inversion layer forms closely to the interface of HfO_2_/Silicon and is extremely thin; the total on-state current only has a small degradation. In addition, Figure 5a,b show that the on-state electron density of CIGAA at the HfO_2_/Silicon interface is much higher than that of GAA, which explains the negligible degradation of the on-state current. Both structures show an increment of off-state current when channel thickness increases, as shown in Figure 4b. The off-state current of CIGAA is about 2 to 5 times lower than that of GAA, which means that CIGAA has the nature to be used to fabricate a low-power device. The good performance of an off-state current can be clearly explained by examining the electron density plot. Figure 6a,b show the electron density of GAA and CIGAA at off-state (V${}_{GS}$ = 0 V), respectively. It is evident that both two structures have almost the same electron density distribution in the channel. However, when we examine the CIGAA, the inner part of the channel where Core–Insulator is located should have the identical electron density distribution with that of GAA if the silicon is not replaced by Core–Insulator, which means that the current path is narrower than that of GAA, resulting in a smaller off-state current.

The characteristics of GAA and CIGAA in terms of subthreshold slope, switching ratio, and drain-induced barrier lowing are shown in Figure 7a–c, respectively. The subthreshold slope of CIGAA always outperforms that of GAA when they have the same T${}_{ox}$. Equation (Equation 1) [30] can perfectly explain the good results of CIGAA. Due to the reduction of off-state current, while I${}_{ON}$ and V${}_{DD}$ remain constant, the subthreshold slope is lowered:(1)$$S_{avg}=\frac{V_{T}-V_{GOFF}}{\log_{10}\frac{I_{T}}{I_{OFF}}}\approx \frac{V_{DD}}{\log_{10}\frac{I_{ON}}{I_{OFF}}}$$

Since the off-state current are lowered, the switching ratio of CIGAA is expected to be lower than that of GAA, as shown in Figure 7b.

### 3.2. Impact of Core–Insulator Diameter and Material on Device Performance

We have set up experiments to further investigate the impact of Core–Insulator diameter and material on device performance. To simplify the experiments, we fixed the nanowire diameter to 8 nm (as shown in Table 2) and the Core–Insulator materials are Si_3_N_4_, SiO_2_, and HfO_2_.

The results in terms of on-state current and off-state current are shown in Figure 8a,b, respectively. Changing of Core–Insulator material only have a minor effect on on-state current based on Figure 8a. In fact, as $D_{CI}$ increases, the impact of Core–Insulator on on-state current become more and more significant. Unlike on-state current, the changing of Core–Insulator material has a conspicuous influence on the off-state current. According to the simulation results (as shown in Figure 8b), SiO_2_ can effectively suppress current flow under off-state, HfO_2_ is the worst material to achieve low off-state current among the three, and Si_3_N_4_ is better than HfO_2_. No matter what the Core–Insulator material is, the on-state current will decrease when $D_{CI}$ increases, since $D_{CI}$ increases means that channel thickness decreases which lead to the reduction of an effective channel width. Likewise, the off-state current will decrease when $D_{CI}$ increases because a larger Core–Insulator helps to suppress off-state current.

As for subthreshold swing, switching ratio and DIBL, increasing $D_{CI}$ results in better performance, as shown in Figure 9a–c, respectively. SiO_2_ can effectively enhance device performance, Si_3_N_4_ is less useful than SiO_2_ and HfO_2_ is the worst choice. In real fabrication, it is important to decide the $D_{CI}$ according to application requirements. SiO_2_ is the best material among the three to achieve better performance. Since larger $D_{CI}$ can reduce on-state current, so the first thing to do is to select $D_{CI}$, which enables on-state current to be large enough.

### 3.3. Impact of Core–Insulator Length on Device Performance

Due to the limitations of existing fabrication technology, it is hard to fabricate a device that has a Core–Insulator exactly the same length as its channel, which is shown in Figure 10. Thus, we have to further investigate the impact of Core–Insulator length on device performance. There are two possible situations: one is that the Core–Insulator extends itself into source and drain by L${}_{ext}$, as shown in Figure 10a. The other is that the Core–Insulator recesses itself into channel by L${}_{ext}$, as shown in Figure 10b.

The results are shown in Table 3. A positive L${}_{ext}$ represents the situation that is shown in Figure 10a, and a negative L${}_{ext}$ represents the situation that is shown in Figure 10b. From the results, we can notice that the performance of CIGAA only has slight variations and can be neglected. When L${}_{ext}$ changes from −2 nm to 2 nm, the Core–Insulator extends itself into source, resulting in a volume reduction of the channel. This reduction causes the effective channel width to be further lowered, which finally results in a reduction in on-state and off-state current. The simulation results reveal that a small variation in fabrication will not cause noticeable performance degradation.

### 3.4. Parasitic Capacitance of CIGAA and GAA

One important concern about the newly proposed structure is its parasitic capacitance, since the presence of Core–Insulator may affect the charge distribution of CIGAA. We have investigated the impact of Core–Insulator materials, channel thickness, and Core–Insulator diameter on the device’s parasitic capacitance. Figure 11a,b show that the changing of Core–Insulator material will significantly affect the device’s parasitic capacitance. When the Core–Insulator material is HfO_2_, the gate capacitance is about three times larger than that of CIGAA with Si_3_N_4_ Core–Insulator, six times larger than that of CIGAA with SiO_2_ Core–Insulator. As channel thickness increases, the gate capacitance will also increase, as shown in Figure 11a. When T${}_{ox}$ is the same, CIGAA with SiO_2_ Core–Insulator shows the smallest gate capacitance, while CIGAA with HfO_2_ Core–Insulator shows the largest gate capacitance. In addition, the gate capacitance of CIGAA with SiO_2_ Core–Insulator and CIGAA with Si_3_N_4_ Core–Insulator are lower than that of GAA. When Core–Insulator diameter increases, the gate capacitance of CIGAA increases simultaneously. HfO_2_ cannot be used as Core–Insulator material because it provides fairly large gate capacitance, while Si_3_N_4_ and SiO_2_ can be used as Core–Insulator material. SiO_2_ is the best material among the three, and it provides the smallest gate capacitance.

## 4. Conclusions

We have studied the device performance of our proposed CIGAA nanowire using 3D TCAD simulation. Due to CIGAA’s lowered off-state current enabled by Core–Insulator, it shows high on-state current, low off-state current, low subthreshold swing, and high switching ratio. CIGAA has the potential to be used to fabricate low-power systems. Thus, the CIGAA nanowire is a promising candidate to extend CMOS scaling roadmap and future low power CMOS devices.

## Figures and Tables

**Figure 1 micromachines-11-00223-f001:**
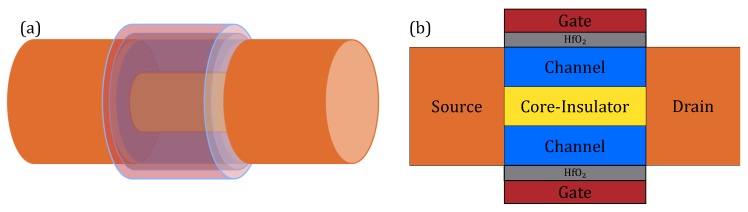
The schematic 3D structure and cross section of CIGAA: (**a**) 3D view of CIGAA; (**b**) cross-sectional view of CIGAA. The illustrations are not depicted proportionally to the really devices; we made some exaggeration for a clear visualization.

**Figure 2 micromachines-11-00223-f002:**
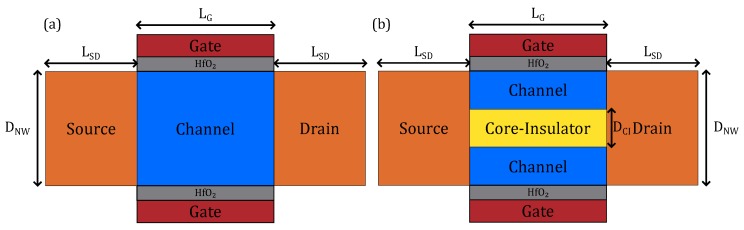
The overall dimension of CIGAA and GAA: (**a**) parameters used for GAA; (**b**) parameters used for CIGAA.

**Figure 3 micromachines-11-00223-f003:**
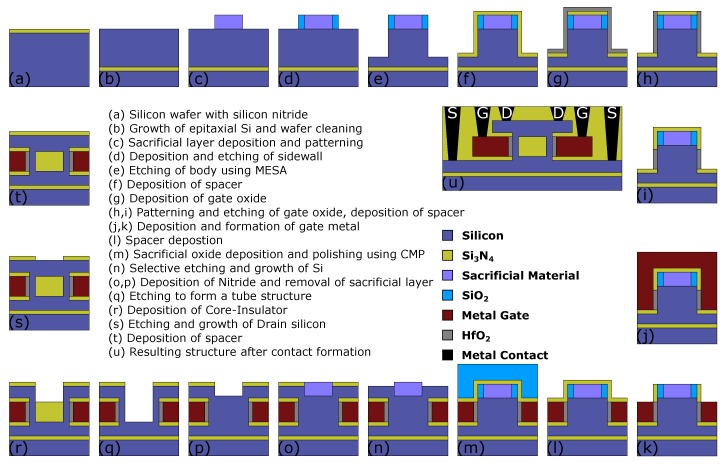
Suggested fabrication process flow for CIGAA.

**Figure 4 micromachines-11-00223-f004:**
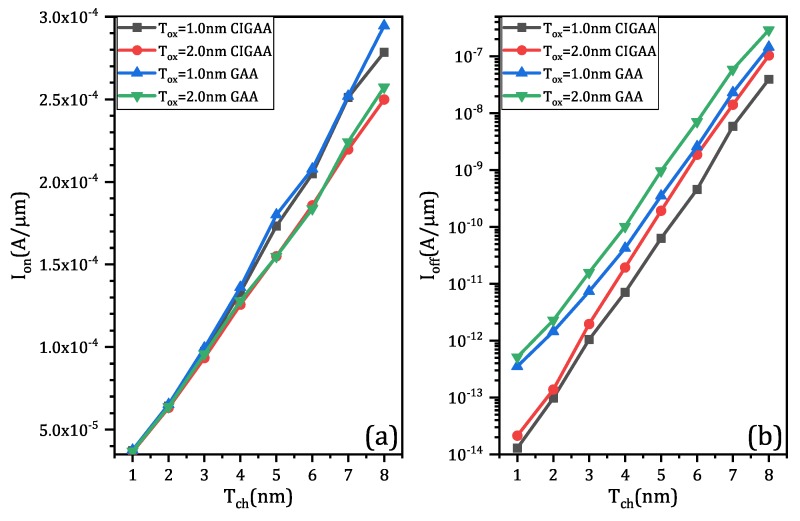
The simulation results of CIGAA and GAA: (**a**) on-state current (I${}_{on}$) of CIGAA and GAA; (**b**) off-state current (I${}_{off}$) of CIGAA and GAA. Both of the two figures are plotted when D${}_{CI}$ = 4.0 nm.

**Figure 5 micromachines-11-00223-f005:**
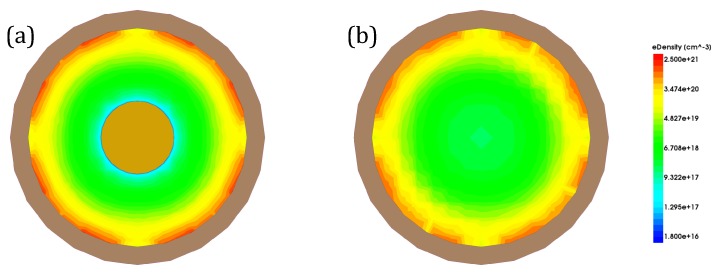
The electron density plot of CIGAA and GAA, the devices are at on state: (**a**) electron density of GAA; (**b**) electron density of CIGAA. Both figures are plotted when V${}_{GS}$ = V${}_{DS}$, V${}_{DS}$ = 1 V. Metal gate and part structures are not included in the figure for clarity.

**Figure 6 micromachines-11-00223-f006:**
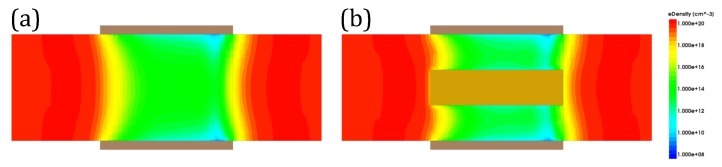
The electron density plot of CIGAA and GAA, the devices are at off state: (**a**) electron density of GAA; (**b**) electron density of CIGAA. Both figures are plotted when V${}_{GS}$ = 0 V, V${}_{DS}$ = 1 V. Metal gate and part structures are not included in the figure for clarity.

**Figure 7 micromachines-11-00223-f007:**
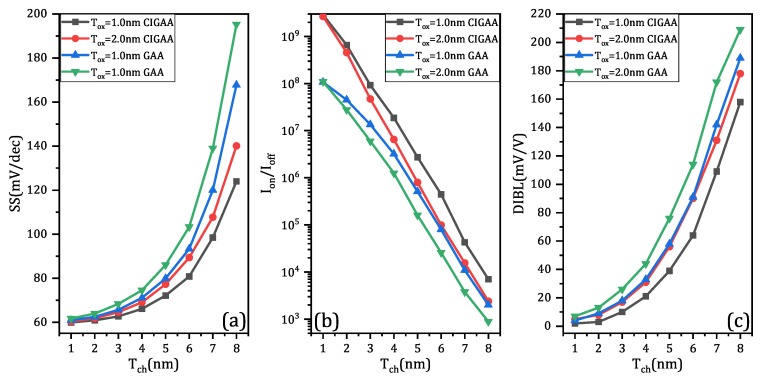
The simulation results of CIGAA and GAA: (**a**) subthreshold slope (SS) of CIGAA and GAA; (**b**) switching ratio (I${}_{on}$/I${}_{off}$) of CIGAA and GAA; (**c**) drain-induced barrier lowering (DIBL) of CIGAA and GAA. All three of the figures are plotted when D${}_{CI}$ = 4.0 nm.

**Figure 8 micromachines-11-00223-f008:**
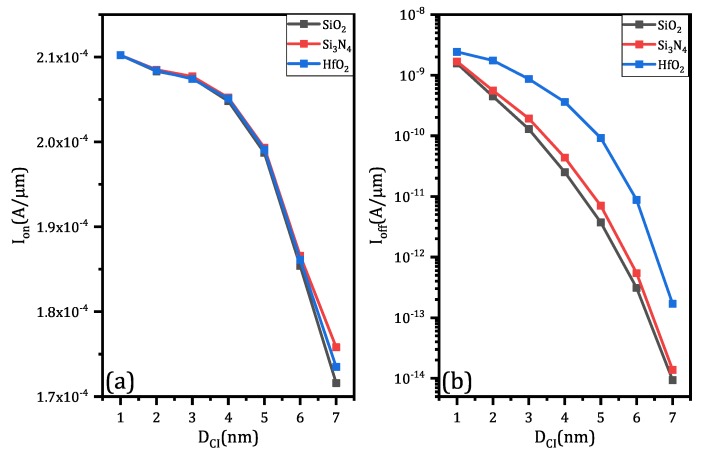
The simulation results of CIGAA and GAA in terms of different Core–Insulator material and D${}_{CI}$: (**a**) on-state current. (**b**) off-state current. Both figures are plotted when D${}_{NW}$ is fixed to 8 nm.

**Figure 9 micromachines-11-00223-f009:**
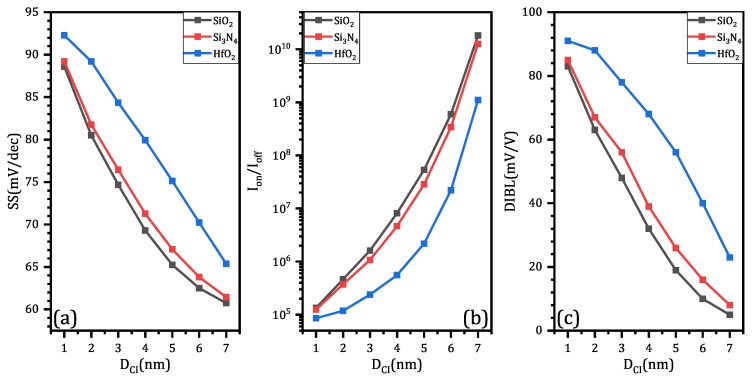
The simulation results of CIGAA and GAA in terms of different Core–Insulator material and D${}_{CI}$: (**a**) subthreshold swing (SS); (**b**) switching ratio (I${}_{on}$/I${}_{off}$); (**c**) drain-induced barrier lowering (DIBL). All three figures are plotted when D${}_{NW}$ is fixed to 8 nm.

**Figure 10 micromachines-11-00223-f010:**
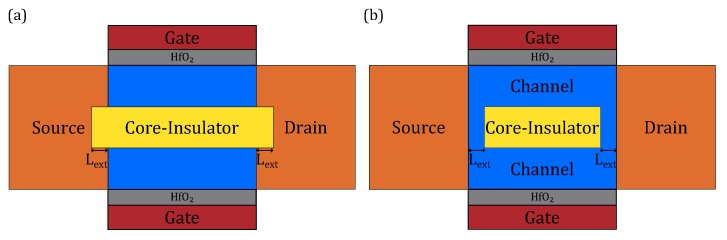
The illustrations which show the issue of Core–Insulator’s extension and contraction: (**a**) the Core–Insulator extends itself into source and drain by L${}_{ext}$; (**b**) the Core–Insulator recesses itself into channel by L${}_{ext}$.

**Figure 11 micromachines-11-00223-f011:**
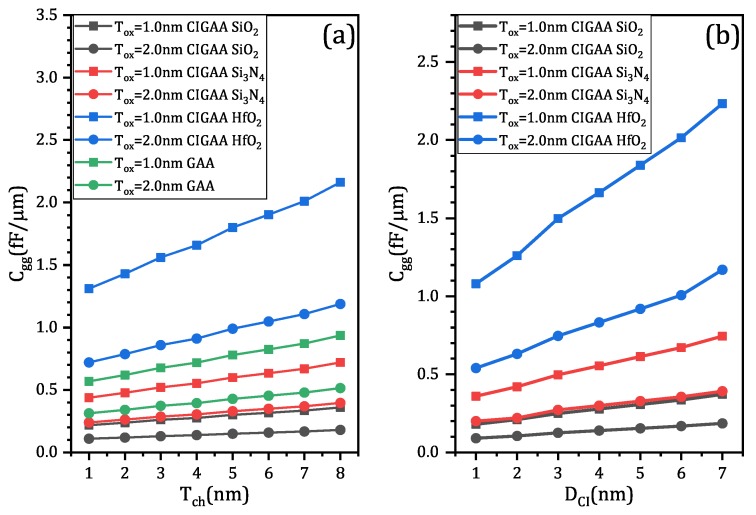
Gate capacitance dependence on: (**a**) channel thickness; (**b**) the Core–Insulator diameter D${}_{CI}$.

**Table 1 micromachines-11-00223-t001:** Design parameter values for CIGAA and GAA.

Variables	Values
Gate Length (L${}_{g}$)	15 nm
HfO_2_ Thickness	1.0 nm/2.0 nm
Channel Thickness (D${}_{NW}$/2D${}_{CI}$/2)	1.0 to 8.0 nm
Source/Drain Length (L${}_{SD}$)	10 nm
Core–Insulator Diameter (D${}_{CI}$)	2 nm to 14 nm
Source/Drain Doping	$1\times 10^{20}$
Channel Doping	$1\times 10^{15}$
Core–Insulator	Si_3_N_4_, SiO_2_, HfO_2_

**Table 2 micromachines-11-00223-t002:** Design parameter values for CIGAA.

Channel Thickness (nm)	Core–Insulator Diameter (nm)
1	7
2	6
3	5
4	4
5	3
6	2
7	1

**Table 3 micromachines-11-00223-t003:** Impact of Core–Insulator Length on Device Performance.

L_*ext*_ (nm)	ΔI_*on*_ (%)	ΔI_*off*_ (%)	ΔI_*on*_/I_*off*_ (%)	ΔSS (%)	ΔDIBL (mV)
−2	0.3	0.5	−0.5	0.7	1
−1	0	0.8	−0.8	0.4	1
0	0	0	0	0	0
1	−0.5	−0.7	0.6	−0.5	0
2	−1.1	−0.4	0.9	−0.7	−1

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
