# Peer review of "A Simulation Study of a Gate-All-Around Nanowire Transistor with a Core–Insulator"

_micromachines, 2020, doi:10.3390/mi11020223_

Round 1

Reviewer 1 Report

The authors propose adding a core insulator to the structure of a gate-all-around MOSFET with the aim of reducing the off current without significantly degrading performance of the device. Overall, the study is sound and there is a good amount of evidence supporting the claim. However there are a few issues that most be answered before granting publication:

Please spend some time improving the language as there are multiple mistakes of syntax (pure language errors) that make it difficult to read. The authors should improve figure 5,  electron density in the channel, to show actual values as no legend is provided. Furthermore, the should provide a slice-cut along the vertical access to support their claim about Ion degradation being less than Ioff despite the channel being reduced. It is of utmost relevance to discuss the effect on parasitic capacitance added with the core insulator. Most likely the frequency response of the device will be significantly reduced making the idea impractical for any useful application. Please add a thorough discussion backed with simulations.

In conclusion this reviewer's opinion is that the manuscript should be reconsider after points 1 to 3 are addressed.

Reviewer 2 Report

The paper is well-organized and the results are interesting.

Comments:

- Revise the manuscript avoiding repetitions of sentences in the different sections (e.g. abstract and introduction).

- The Introduction should be better addressed to the role of a rigorous modelling analysis to support the optimized design of non-conventional MOSFETs in a short time. You can cite the following papers:

[x] J. Singh, M. J. Kumar, “A Planar Junctionless FET Using SiC With Reduced Impact of Interface Traps: Proposal and Analysis”, IEEE Trans. Electron Dev., vol. 64, p. 4430-4434, 2017.

[x] F. Pezzimenti, H. Bencherif, A. Yousfi and L. Dehimi, “Current-voltage analytical model and multiobjective optimization of design of a short channel gate-all-around-junctionless MOSFET,” Solid-State Electron., vol. 161, p. 107642, 2019.

[x] F. Pezzimenti, “Modeling of the steady state and switching characteristics of a normally-off 4H-SiC trench bipolar-mode FET” IEEE Transactions on Electron Devices, vol. 60, p. 1404-1411, 2013

[x] M.G. Jaikumar, S. Karmalkar, “Calibration of Mobility and Interface Trap Parameters for High Temperature TCAD Simulation of 4H-SiC VDMOSFETs”, Material Science Forum, vol. 717-720, p. 1101-1104, 2012.

- In Figs. 4, 6, 7, and 8, add labels (a, b, c, etc.) to the figures themselves.  

- Report in the text some experimental results in terms of Ion and Ioff for comparison purposes.
